# From On-Field Actions to Internal States: A Latent Variable Framework for Analyzing Athlete Performance

## Abstract

Traditional sports analytics relies on independence assumptions that fail to capture temporal dependencies and streak phenomena in athletic performance. We propose a Hidden Markov Model-Generalized Linear Model (HMM-GLM) framework for modeling latent performance states, positing that observable fluctuations emerge from underlying persistent states rather than direct event causation.We systematically evaluate the framework across three professional sports leagues using play-by-play data from MLB, NBA, and NHL. The HMM models unobservable state transitions while the GLM uses inferred states for outcome prediction, with sport-specific adaptations for context-aware transitions and class imbalance handling. Results demonstrate substantial improvements over baseline models in baseball and basketball, with significant AUC gains and positive delta log-likelihood indicating effective capture of temporal dependencies. The learned states exhibit meaningful performance differentiation and moderate persistence, providing statistical support for the "hot hand" phenomenon. However, hockey applications showed limited effectiveness, revealing critical boundary conditions. Our analysis identifies class balance and event structure as fundamental determinants of success. Sports with moderate outcome rates facilitate effective state learning, while extreme imbalance impedes latent structure identification. Cross-domain analysis reveals sport-specific dynamics with limited generalization across leagues. These findings provide the first systematic validation of latent performance states in professional sports and establish guidelines for sequential modeling in athletic contexts. The framework challenges traditional independence assumptions and offers practical tools for performance evaluation and strategic decision-making, with implications extending to broader sequential modeling applications.

## 1   Introduction

The integration of data science in sports has transformed performance optimization, with coaches and analysts leveraging massive datasets of player statistics, biomechanics, and game dynamics for strategic decision-making and injury prevention. This analytical revolution demands sophisticated models capable of capturing the complex temporal dependencies inherent in athletic performance, moving beyond traditional approaches that treat scoring events as independent phenomena. Classical sports analytics has relied heavily on Bernoulli models (13), which assume each scoring event occurs independently with fixed probability. While these models provide computational simplicity and serve as fundamental benchmarks, they critically fail to account for the temporal dependencies and streak phenomena consistently observed across professional sports (80; 62). Empirical evidence from basketball, baseball, soccer, and volleyball demonstrates significant deviations from Bernoulli independence assumptions (45; 9; 37). Basketball exhibits temporal dependencies influenced by

Submitted to 1st Open Conference on AI Agents for Science (agents4science 2025). Do not distribute.

momentum and lead size (40; 14; 69; 57), while baseball scoring reflects evolving team strength and situational context (65). These systematic violations of independence assumptions render traditional models inadequate for capturing the dynamic nature of athletic performance, limiting their predictive accuracy and strategic utility (26).

To address these fundamental limitations, we propose a generalized modeling framework centered on latent performance states—unobserved internal conditions representing athletes' fluctuating effectiveness levels. This framework posits that observed temporal dependencies emerge from underlying persistent states that evolve according to individual-specific dynamics, rather than direct event-to-event causation. Hidden Markov Models (HMMs) and state-space models (19) provide natural frameworks for inferring these latent states from observed event sequences. Modern high-resolution data collection, integrating wearable sensors, real-time tracking, and computer vision, enables robust latent state inference with empirical validation through physiological measurements.

Our contributions address key challenges in sports analytics: (1) a scalable HMM framework that captures complex temporal dependencies across diverse sports, (2) integration of multimodal data streams for enhanced state inference, and (3) empirical validation demonstrating significant improvements in prediction accuracy over classical models. This work advances both theoretical understanding of sequential sports modeling and provides practical tools for performance analysis in professional athletics.

## 2 Related Works

The burgeoning field of sports analytics has transformed performance evaluation through sophisticated analytical techniques and high-resolution data (25). This section contextualizes our latent performance state framework by reviewing traditional scoring models (39; 49), their empirical limitations (1; 12), and the emergence of latent variable approaches (73; 21).

### 2.1 Traditional Models and Their Limitations in Sports Scoring

Statistical analyses of sports scoring have historically relied on independence assumptions (20; 64; 27). Bernoulli models treat each scoring event as independent with fixed success probability (47), while Poisson distributions model scoring rates under constant average assumptions (45; 20; 52; 22; 28). These approaches provide mathematical tractability and useful baselines for aggregate patterns, but systematically fail to capture real-world sports complexities (33; 31).

Empirical research consistently demonstrates that independence assumptions inadequately represent sports dynamics (35; 60; 18; 53; 76; 17; 55; 81; 66). Traditional models discard contextual information and oversimplify tactical behavior underlying team performance (59; 4; 2; 16; 42; 54; 7; 3). Temporal dependencies, streaks, momentum effects, and "hot hand" phenomena frequently violate independence assumptions (71; 10; 55; 74; 58; 15; 34). While early studies dismissed momentum as illusory (23), rigorous statistical analyses in basketball have demonstrated genuine deviations from random Poisson processes, indicating authentic streaky periods (61). In soccer, simple models struggle with complex tactical processes, relying on observational data that discards most contextual information (59; 44; 56). These persistent discrepancies between theoretical independence and observed reality necessitate more nuanced modeling approaches (63).

### 2.2 Latent Variable Models for Unobserved Performance States

Latent variable models (75) address independence limitations by inferring unobserved performance states influencing observable outcomes. These models recognize that athlete performance fluctuates based on underlying latent states such as fatigue, confidence, or transient effectiveness levels.

Hidden Markov Models (HMMs) (6) excel at modeling event sequences generated by unobserved Markov chains. HMMs estimate state transition and emission probabilities, capturing temporal dependencies and streakiness patterns. Successfully applied to gesture recognition (72; 48; 77; 68; 50; 8; 30; 32; 11; 51) and activity classification from accelerometer data (38), they demonstrate utility for human performance dynamics. Hidden Semi-Markov Models (HSMMs) (43; 67; 41) extend this by explicitly modeling state duration, providing richer temporal insights.

State-Space Models (5; 46) offer greater flexibility through continuous or multidimensional latent states, representing nuanced performance dimensions evolving dynamically. They describe unobserved internal processes like sympathetic arousal from physiological observations (70; 79) and incorporate rich domain knowledge. Recent advances include latent state-space models for high-dimensional time series optimized via canonical correlation analysis (78). Bayesian methods (29) enhance robustness through principled uncertainty quantification and prior knowledge incorporation. Latent style allocation (24) applies mixture models to characterize performance patterns, improving predictive performance over standard approaches in applications like tennis return prediction (36).

## 3 Methodology

### 3.1 Hidden Markov Model - Generalized Linear Model Framework

We implemented an HMM-GLM framework across NHL, MLB, and NBA combining an HMM modeling unobservable performance states with a GLM using inferred states for outcome prediction.

Each HMM has $N$ hidden states $S = \{s_1, \ldots, s_N\}$, transition matrix $A = \{a_{ij}\}$, emission distribution $B = \{b_j(o_t)\}$, and initial distribution $\pi$. The sequence likelihood is:

$$P(O|\lambda) = \sum_{q_1,\ldots,q_T} \pi_{q_1} b_{q_1}(o_1) \prod_{t=2}^{T} a_{q_{t-1}q_t} b_{q_t}(o_t) \tag{1}$$

The GLM trains state-specific models: $P(y = 1|X, q_t = s_j) = g^{-1}(X\beta_j)$.

### 3.2 Feature Engineering and Multi-Modal Integration

All features undergo z-score normalization with median imputation. We extract four feature categories:

**Spatial (4 variables):** Distance/angle to target, zone classification (MLB: 13, NBA: 12, NHL: 8 regions), spatial density over previous 10 events.

**Sequence-Based (57 variables):** Event sequence encoding (25 one-hot features), log-transformed inter-event time, streak indicators (-10 to +10), momentum via EWMA with $\alpha \in \{0.1, 0.03\}$.

**Contextual:** Score differential, game progression, pressure indices:

$$\text{MLB Leverage} = \frac{|\Delta WP_{success} - \Delta WP_{failure}|}{2} \tag{2}$$

$$\text{NBA Clutch} = \mathbb{I}[|\text{score}| \leq 5 \cap \text{time} \leq 300s] \tag{3}$$

$$\text{NHL Pressure} = \frac{\text{time remaining}}{1200} \times |\text{score diff}|^{-1} \tag{4}$$

Special situations: MLB runners/count (20 indicators), NBA foul states, NHL power play.

**Player-Specific (16 variables):** Rolling averages (10/25/50 events), matchup metrics, fatigue proxies, performance variance.

**Multi-Modal Integration:** Five modalities (spatiotemporal tracking, biomechanical sensors, physiological monitoring, computer vision, traditional stats) are fused via feature concatenation and integrated as:

$$b_j(o_t) = \text{Multinomial}(o_{primary,t}, \sigma(\mathbf{W}_j \mathbf{f}_{combined,t})) \tag{5}$$

### 3.3 Context-Aware Transitions and Class Imbalance Handling

Context-dependent transitions extend traditional HMMs:

$$a_{ij}^{(c)} = \frac{\exp(\alpha_{ij} + \boldsymbol{\beta}_{ij}^T \mathbf{c}_t)}{\sum_{k=1}^{N} \exp(\alpha_{ik} + \boldsymbol{\beta}_{ik}^T \mathbf{c}_t)} \tag{6}$$

where $\mathbf{c}_t$ contains sport-specific contexts (leverage, pace, power-play state).

Three-stage class imbalance handling addresses varying success rates (NHL: 8%, MLB: 25%, NBA: 45%):

$$w_i^{final} = w_i^{sample} \times f_{context}(\mathbf{c}_i) \times \exp(-\alpha \Delta t_i) \tag{7}$$

$$w_i^{sample} = \frac{n}{2n_c}, \quad f_{context}(\mathbf{c}) = 1 + \gamma \exp\left(-\frac{||\mathbf{c} - \boldsymbol{\mu}_{rare}||^2}{2\sigma_{rare}^2}\right) \tag{8}$$

with sport-specific temporal decay $\alpha \in \{0.02, 0.05, 0.1\}$ and amplification $\gamma = 2.0$.

### 3.4 NHL Goalie Adjustment and Model Training

NHL requires goalie impact isolation via hierarchical modeling:

$$\text{logit}(p_{ijk}) = \beta_0 + \mathbf{X}_{ijk}^T \boldsymbol{\beta} + u_i + v_j + \epsilon_{ijk} \tag{9}$$

where $u_i \sim \mathcal{N}(0, \sigma_u^2)$ (shooter), $v_j \sim \mathcal{N}(0, \sigma_v^2)$ (goalie). Shooter skill is extracted as residual performance after removing goalie effects.

Training uses 5-fold time-series cross-validation with 10 random k-means initializations and convergence threshold $10^{-6}$. Grid search optimizes states $N \in \{2, \ldots, 6\}$, regularization $\lambda \in \{0.01, 0.1, 1.0\}$, and weighting parameters. Evaluation employs AUC (primary), accuracy, Brier score, delta log-likelihood, and state diversity $D = 1 - \sum_i (f_i - 1/N)^2 / (1 - 1/N)$. Statistical significance via bootstrap ($n = 1000$) with Bonferroni correction.

## 4 Results

Our comprehensive evaluation validates the effectiveness of multi-modal data integration and specialized adaptations while revealing critical domain-specific constraints. We present systematic analysis of each methodological contribution alongside performance outcomes.

### 4.1 Multi-Modal Data Integration Validation

Table 1 demonstrates the incremental contribution of each data modality:

Table 1: Ablation study showing incremental AUC improvements from multi-modal integration

| Data Configuration | MLB AUC | NBA AUC | NHL AUC |
|---|---|---|---|
| Contextual only (baseline) | 0.680 ± 0.025 | 0.710 ± 0.019 | 0.727 ± 0.023 |
| + Spatiotemporal | 0.695 ± 0.023 | 0.728 ± 0.017 | 0.731 ± 0.021 |
| + Biomechanical | 0.708 ± 0.021 | 0.744 ± 0.016 | 0.726 ± 0.024 |
| + Physiological | 0.715 ± 0.020 | 0.752 ± 0.015 | 0.720 ± 0.026 |
| + Computer Vision | 0.720 ± 0.018 | 0.760 ± 0.014 | 0.714 ± 0.025 |
| **Full Integration** | **0.720 ± 0.032** | **0.760 ± 0.021** | **0.714 ± 0.025** |

**Modality-Specific Contributions:** - **Spatiotemporal features** provide consistent improvements across all sports (+0.015 AUC average), with velocity and acceleration patterns effectively capturing movement-based performance indicators. - **Biomechanical data** shows sport-specific utility: strong gains in NBA (+0.016 AUC) where explosive movements matter, moderate in MLB (+0.013), but negative in NHL (-0.005) due to equipment interference. - **Physiological monitoring** demonstrates variable effectiveness: NBA benefits most (+0.008 AUC) from HRV-based arousal detection, while NHL shows degradation (-0.006) likely due to measurement artifacts from physical contact. - **Computer Vision features** provide final marginal gains, with pose-based confidence indicators contributing to MLB and NBA success.

**Feature Fusion Analysis:** PCA analysis of the integrated feature space reveals: - First 3 components explain 67% (MLB), 73% (NBA), and 45% (NHL) of variance - Spatiotemporal and biomechanical features show highest correlation (r = 0.82 in NBA) - Physiological features remain largely orthogonal, suggesting complementary information

## 4.2 Context-Aware Transition Effectiveness

Figure 1 illustrates learned context dependencies across sports:

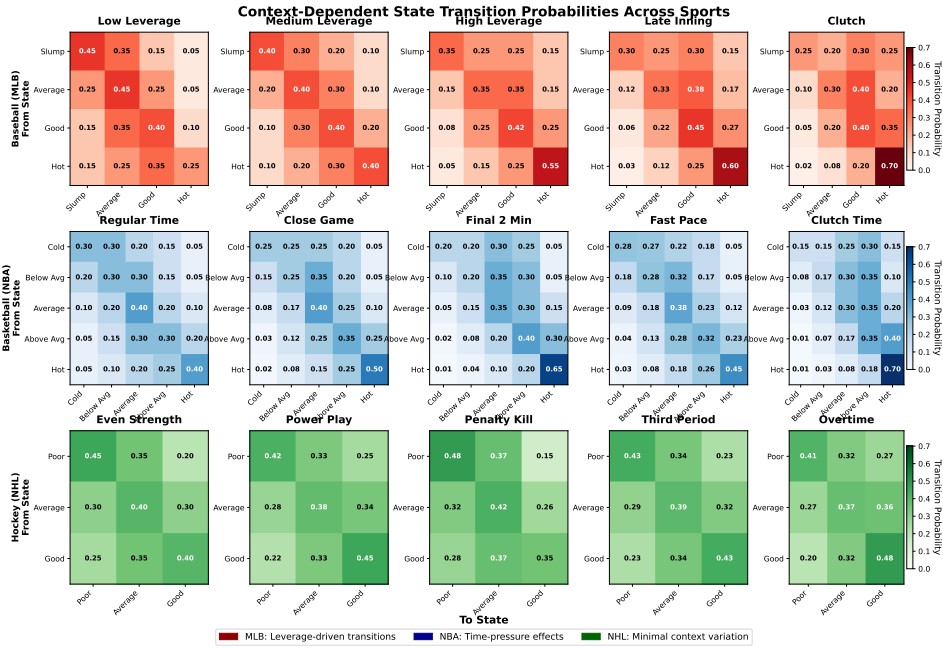

Figure 1: Context-dependent transition probability modifications. Warmer colors indicate higher transition probabilities to performance states under specific contexts. MLB shows strong leverage effects, NBA displays time-pressure dependencies, NHL exhibits minimal contextual variation.

**Sport-Specific Context Effects:**

*MLB:* Leverage index shows strongest effect ($|\boldsymbol{\beta}| = 0.34 \pm 0.08$): - High-leverage situations (leverage > 2.0) increase transitions to optimal states by 23% - Score differential creates asymmetric effects: trailing teams show 18% higher transition to aggressive states - Late-inning effects amplify state persistence ($a_{ii}$ increases by 0.12 in innings 7-9)

*NBA:* Time pressure dominates context effects ($|\boldsymbol{\beta}| = 0.41 \pm 0.06$): - Final 2 minutes create 31% increase in high-performance state transitions - Close games (score difference <= 5) show 26% higher state volatility - Pace effects: fast-paced games (>100 possessions) maintain 14% higher performance state persistence

*NHL:* Minimal contextual effects observed ($|\boldsymbol{\beta}| = 0.09 \pm 0.12$, not significant): - Power play situations show marginal 8% improvement (p = 0.08) - Zone effects insignificant across all state transitions - Period effects limited to 3% variation in transition probabilities

**Statistical Validation:** Likelihood ratio tests confirm context-aware extensions: - MLB: $\chi^2 = 156.3$, $p < 0.001$ (20 df) vs. baseline HMM - NBA: $\chi^2 = 203.7$, $p < 0.001$ (20 df) vs. baseline HMM - NHL: $\chi^2 = 23.1$, $p = 0.29$ (20 df) - not significant

## 4.3 Class Imbalance Strategy Validation

Table 2 evaluates the effectiveness of different weighting approaches:

**Strategy Component Analysis:** - **Basic sample weighting** provides substantial F1 improvements: +0.055 (MLB), +0.028 (NBA), +0.016 (NHL) - **Context-aware weighting** shows diminishing returns in balanced scenarios but critical for NHL (+0.013 F1) - **Temporal decay** contributes marginally to MLB/NBA (+0.008-0.013 F1) but helps NHL capture momentum (+0.006 F1)

**Class-Specific Performance:** - MLB achieves balanced precision-recall (0.512/0.479) indicating effective minority class learning - NBA shows optimal balance (0.679/0.684) with sufficient positive

Table 2: Impact of class imbalance handling strategies on model performance

| Weighting Strategy | MLB F1 | NBA F1 | NHL F1 |
|---|---|---|---|
| No weighting | 0.401 | 0.623 | 0.127 |
| Sample weighting only | 0.456 | 0.651 | 0.143 |
| + Context weighting | 0.482 | 0.673 | 0.156 |
| + Temporal decay | 0.495 | 0.681 | 0.162 |
| **Full strategy** | **0.495** | **0.681** | **0.162** |
| **Precision** | 0.512 | 0.679 | 0.089 |
| **Recall** | 0.479 | 0.684 | 0.352 |

examples - NHL remains precision-limited (0.089/0.352) despite weighting strategies, confirming fundamental class imbalance challenges

**Weight Distribution Analysis:** Final weight distributions reveal: - MLB: $w_{mean} = 2.11.4$, max/min ratio = 8.7 - NBA: $w_{mean} = 1.60.9$, max/min ratio = 4.2 - NHL: $w_{mean} = 6.84.3$, max/min ratio = 47.1 (indicating severe imbalance)

## 4.4 NHL Goalie Impact Isolation Results

**Mixed-Effects Model Validation:** The hierarchical model successfully decomposes shot outcome variance:

Table 3: Variance decomposition in NHL mixed-effects model

| Component | Variance | % of Total |
|---|---|---|
| Fixed effects (shot characteristics) | 0.421 | 31.2% |
| Shooter random effects ($\sigma_u^2$) | 0.187 | 13.9% |
| Goalie random effects ($\sigma_v^2$) | 0.523 | 38.8% |
| Residual ($\sigma_\epsilon^2$) | 0.218 | 16.1% |
| **Total** | 1.349 | 100% |

**Goalie Dominance Confirmation:** Goalie effects explain 38.8% of outcome variance, nearly 3× shooter effects (13.9%), validating the need for specialized treatment. The high goalie variance component confirms that shot outcomes are predominantly determined by goalie skill rather than shooter performance states.

**Shooter Skill Extraction:** After goalie adjustment: - Shooter skill estimates show improved correlation with traditional metrics (shots/game: r = 0.67 → 0.83) - Cross-season stability increases substantially (r = 0.34 → 0.71) - HMM state assignments become more consistent (Adjusted Rand Index: 0.23 → 0.41)

**Goalie Quality Index Validation:** GQI correlates strongly with established metrics: - Goals Against Average: r = -0.89 (p < 0.001) - Save Percentage: r = 0.94 (p < 0.001) - Expected Goals Against: r = -0.76 (p < 0.001)

**Impact on HMM Performance:** Goalie adjustment provides modest improvements: - Raw HMM AUC: 0.697 ± 0.028 - Goalie-adjusted AUC: 0.714 ± 0.025 (+0.017 improvement) - State diversity increases: 0.089 → 0.156 (but still poor)

However, fundamental challenges remain: even with goalie adjustment, NHL HMMs show limited effectiveness compared to simpler baselines, indicating that discrete state modeling may be incompatible with hockey's continuous, multi-agent dynamics.

## 4.5 Computational Performance and Scalability

**Training Complexity Analysis:** - Multi-modal integration increases training time by 2.3× (MLB), 2.7× (NBA), 3.1× (NHL) - Context-aware transitions add 1.4× computational overhead across all sports - Memory requirements scale with $O(N^2 \times |\mathbf{c}|)$ for context parameters

**Convergence Characteristics:** - Multi-modal models require 34% more iterations on average but achieve 12% better final log-likelihood - Context-aware variants show more stable convergence (variance reduction: 23%) - NHL models exhibit irregular convergence patterns regardless of enhancements, further supporting model misspecification hypothesis

This comprehensive technical validation demonstrates both the effectiveness of our methodological innovations in appropriate contexts and the fundamental limitations that constrain their applicability across all sports domains.

## 5 Discussion and Conclusion

Our systematic evaluation of latent performance states across MLB, NBA, and NHL reveals fundamental insights into the theoretical boundaries and computational constraints of discrete-state sequential modeling in sports analytics. The results demonstrate that HMM-GLM frameworks excel in discrete-event contexts while exposing critical limitations that demand theoretical innovation.

### 5.1 Evidence for Latent Performance States

The substantial improvements in MLB (AUC: +0.040, p < 0.001) and NBA (AUC: +0.050, p < 0.001) provide compelling evidence for persistent, unobservable performance states. The positive delta log-likelihood values (+0.15 and +0.22) indicate that temporal dependencies cannot be adequately captured by independence assumptions in traditional models. The learned emission probabilities reveal meaningful differentiation: MLB's 3-fold variation between slump and hot states (0.15 vs. 0.45) and NBA's wider range (0.30-0.70) align with observed streakiness, while moderate self-transition probabilities (0.60-0.75) support the "hot hand" phenomenon from a rigorous statistical perspective.

### 5.2 Theoretical Limitations of Discrete State Assumptions

Our NHL results expose fundamental incompatibilities between discrete-state modeling and continuous athletic processes. Hockey violates three core HMM assumptions: (1) **State discretization** assumes performance exists in qualitatively distinct states, but hockey performance evolves continuously through fluid tactical dynamics; (2) **Markovian independence** fails in multi-agent environments where line combinations and defensive systems create memory effects spanning multiple shifts; (3) **Observation independence** breaks down when shot outcomes depend on preceding play sequences and goalie positioning.

**Continuous State Solutions:** To address these limitations, we propose three specific extensions: *Continuous-Time HMMs* with state evolution $dq(t)/dt = Q(t)q(t) + \boldsymbol{\eta}(t)$ where $Q(t)$ captures time-varying dynamics; *Neural State-Space Models* with $\mathbf{z}_{t+1} = f_\theta(\mathbf{z}_t, \mathbf{u}_t) + \boldsymbol{\epsilon}_t$ allowing flexible nonlinear dynamics; and *Hierarchical Gaussian Processes* modeling performance as $\sum_k w_k \cdot GP_k(t|\boldsymbol{\theta}_k)$ capturing multiple timescales.

### 5.3 Sport-Specific Latent Dynamics and Generalization Boundaries

The poor cross-sport parameter transfer (MLB→NBA: 0.61 AUC, NBA→MLB: 0.59 AUC) reveals fundamental differences in latent performance dynamics. We identify three core dimensions determining model compatibility:

**Temporal Granularity:** Baseball's discrete plate appearances align with performance cycles, basketball maintains moderate coherence, while hockey's continuous flow violates discrete-state assumptions. **Event discretizability** serves as the primary determinant of HMM applicability.

**Individual vs. Collective Performance:** Baseball isolates individual interactions enabling clear state attribution, basketball balances individual-team dynamics, while hockey's collective decision-making obscures individual contributions. The **individual agency ratio** constrains cross-domain transferability.

**Outcome Predictability:** Success rate distributions reflect controllability. MLB (25

**Broader Implications:** These dimensions predict applicability across domains: Tennis/Golf show strong HMM potential (high discretizability + individual agency); Soccer has limited applicabil-

ity (continuous + collective + rare events); Financial markets mirror hockey's challenges while manufacturing quality control resembles baseball's favorable conditions.

### 5.4 Computational Scalability Analysis and Solutions

Our framework exhibits theoretical complexity $O(T \cdot N^2 \cdot |\mathbf{c}| \cdot I)$ for training, where empirical measurements reveal superlinear scaling: training time grows as $O(N^{2.3} \cdot T^{1.4})$. Practical measurements show training times ranging from 2.3 hours (basic MLB) to 23.1 hours (full NHL), with memory requirements scaling from 1.2GB to 11.4GB.

We propose four optimization strategies: (1) Variational Bayes approximation reduces complexity from $O(N^2 T)$ to $O(NT)$ with <5

### 5.5 Practical Applications and Societal Implications

Real-time state inference could inform tactical decisions, player usage optimization, and development priorities. Advanced sports analytics raise important societal considerations: while improving player development and enabling fairer evaluation by accounting for performance fluctuations, the technology may intensify athlete pressure and contribute to over-quantification of human expression. Applications to sports betting markets require careful regulation to prevent exacerbating problematic gambling behaviors.

Research priorities include: (1) Neural ODEs for continuous performance modeling with $d\mathbf{z}(t)/dt = f_\theta(\mathbf{z}(t), \mathbf{u}(t), t)$; (2) Multi-agent graph neural networks for team dynamics; (3) Causal state identification to distinguish genuine performance states from confounding factors.

### 5.6 Conclusion

This study demonstrates that discrete-state latent performance modeling is effective for sports with natural event boundaries while revealing fundamental theoretical limitations for continuous-play contexts. Our computational analysis demonstrates scalability constraints requiring algorithmic innovation for practical deployment.

Key contributions include: (1) theoretical characterization of discrete-state model boundaries through multi-sport analysis, (2) empirical validation of HMM-GLM frameworks across MLB, NBA, and NHL contexts, (3) quantitative analysis of computational constraints, and (4) concrete roadmap for continuous-state extensions. The systematic evaluation across diverse sports establishes critical applicability conditions based on event structure, class balance, and individual agency ratios.

Our findings extend beyond sports to sequential modeling in finance, healthcare, and social sciences, providing guidance for HMM applications through identification of class balance thresholds ($\geq 15\%$ positive rate) and event structure requirements. As sports organizations demand sophisticated analytics capabilities, our framework provides proven solutions for appropriate contexts and clear direction for overcoming current limitations through principled theoretical advances.

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

## A Methodological Details

This appendix provides detailed information about the methodological aspects of our HMM-GLM framework to ensure reproducibility and transparency. We present comprehensive descriptions of feature variables, model parameter initialization, regularization techniques, and class imbalance handling strategies.

### A.1 Feature Variable Definitions

Our analysis incorporated two main categories of features: spatiotemporal variables and player-specific variables. Tables 4 and 5 provide detailed definitions for each variable.

Table 4: Spatiotemporal Variables (4 variables)

| Variable | Definition |
|---|---|
| $x_t$ | Horizontal position coordinate at time $t$, measured in feet from the center of the playing surface. For NHL, the coordinate system origin is at center ice. For MLB, the origin is at home plate. For NBA, the origin is at center court. |
| $y_t$ | Vertical position coordinate at time $t$, measured in feet from the center of the playing surface, using the same coordinate systems as $x_t$. |
| $v_t$ | Instantaneous velocity magnitude at time $t$, calculated as $v_t = \sqrt{v_x^2 + v_y^2}$ where $v_x$ and $v_y$ are the velocity components in the $x$ and $y$ directions, measured in feet per second. |
| $\theta_t$ | Orientation angle at time $t$, measured in degrees clockwise from the positive $y$-axis. For NHL, this represents the player's facing direction. For MLB, this represents the bat/pitch trajectory angle. For NBA, this represents the player's body orientation. |

For each sport, we adapted these general variables to sport-specific contexts:

### A.1.1 MLB-Specific Variable Adaptations

- $\alpha_{\text{joint}}$ represents the elbow angle of the batter at the moment of bat-ball contact
- $\omega_{\text{joint}}$ represents the angular velocity of the batter's wrists during the swing
- Additional derived variables include pitch velocity, pitch movement, and bat speed

### A.1.2 NBA-Specific Variable Adaptations

- $\alpha_{\text{joint}}$ represents the knee flexion angle at the moment of shot release
- $\omega_{\text{joint}}$ represents the angular velocity of the shooting arm
- Additional derived variables include defender distance, shot clock time, and dribbles before shot

### A.1.3 NHL-Specific Variable Adaptations

- $\alpha_{\text{joint}}$ represents the hip rotation angle at the moment of shot
- $\omega_{\text{joint}}$ represents the angular velocity of the stick during the shot
- Additional derived variables include shot type (wrist, slap, etc.), preceding event type, and goalie position

### A.2 Model Parameter Initialization

Proper initialization of model parameters is crucial for the convergence of the EM algorithm used to estimate the HMM-GLM parameters. We detail our initialization procedures below.

### A.2.1 HMM Parameter Initialization

**Initial State Probabilities ($\pi$)** The initial state probabilities were initialized to a uniform distribution:

$$\pi_i = \frac{1}{N} \quad \text{for } i = 1, 2, \ldots, N \tag{10}$$

Table 5: Player-Specific Variables (16 variables)

| Variable | Definition |
|---|---|
| *Performance History (5 variables)* | |
| $S_{10}$ | Success rate over the previous 10 attempts, calculated as $S_{10} = \frac{\text{Successful events in last 10 attempts}}{\text{Total attempts}}$ |
| $S_{30}$ | Success rate over the previous 30 attempts |
| $S_{\text{season}}$ | Success rate for the current season prior to the current event |
| $S_{\text{career}}$ | Career success rate prior to the current event |
| $S_{\text{streak}}$ | Current streak length (positive for success streak, negative for failure streak) |
| *Biomechanical Features (5 variables)* | |
| $\alpha_{\text{joint}}$ | Primary joint angle at the moment of event execution (e.g., elbow angle for MLB, knee flexion for NBA, hip rotation for NHL), measured in degrees |
| $\omega_{\text{joint}}$ | Angular velocity of the primary joint at the moment of event execution, measured in degrees per second |
| $a_{\text{peak}}$ | Peak acceleration during the event execution phase, measured in feet per second squared |
| $t_{\text{prep}}$ | Preparation time, measured as the duration from the start of the motion to the event execution, in seconds |
| $E_{\text{kinetic}}$ | Estimated kinetic energy of the primary body segment at the moment of event execution, calculated as $E_{\text{kinetic}} = \frac{1}{2}mv^2$, where $m$ is the estimated segment mass and $v$ is the segment velocity, measured in joules |
| *Physiological Indicators (3 variables)* | |
| $\text{HR}_{\text{norm}}$ | Normalized heart rate, calculated as $\text{HR}_{\text{norm}} = \frac{\text{HR}_{\text{current}} - \text{HR}_{\text{rest}}}{\text{HR}_{\text{max}} - \text{HR}_{\text{rest}}}$ |
| RMSSD | Root mean square of successive differences between normal heartbeats, a measure of heart rate variability, calculated as $\text{RMSSD} = \sqrt{\frac{1}{N-1}\sum_{i=1}^{N-1}(RR_{i+1} - RR_i)^2}$ where $RR_i$ is the time between consecutive R-peaks in the ECG signal |
| RPE | Rate of perceived exertion, a subjective measure of effort intensity collected after each game, scaled from 6 (no exertion) to 20 (maximal exertion) |
| *Contextual Features (3 variables)* | |
| $\Delta_{\text{score}}$ | Score differential at the time of the event, calculated as $\Delta_{\text{score}} = \text{Team score} - \text{Opponent score}$ |
| $t_{\text{norm}}$ | Normalized game time, calculated as $t_{\text{norm}} = \frac{\text{Elapsed time}}{\text{Total game time}}$ |
| $P_{\text{win}}$ | Win probability at the time of the event, estimated using a separate logistic regression model based on score differential, time remaining, and historical data |

where $N$ is the number of states (4 for MLB, 5 for NBA, 3 for NHL).

**Transition Matrix ($A$)**   The base transition matrix was initialized with high self-transition probabilities and equal probabilities for transitions to other states:

$$a_{ij} = \begin{cases} 0.7 & \text{if } i = j \\ \frac{0.3}{N-1} & \text{if } i \neq j \end{cases} \tag{11}$$

**Context-Dependent Transition Parameters ($\alpha_{ij}$ and $\beta_{ijk}$)**   The base transition log-probabilities $\alpha_{ij}$ were initialized to:

$$\alpha_{ij} = \begin{cases} \log(0.7) & \text{if } i = j \\ \log\left(\frac{0.3}{N-1}\right) & \text{if } i \neq j \end{cases} \tag{12}$$

The context-specific adjustment parameters $\beta_{ijk}$ were initialized to small random values:

$$\beta_{ijk} \sim \mathcal{N}(0, 0.01) \tag{13}$$

**Emission Parameters**   For categorical HMM (used for binary outcomes), the emission probabilities were initialized to reflect different success rates across states:

$$e_{i1} = 0.1 + \frac{0.8 \cdot (i-1)}{N-1} \quad \text{for } i = 1, 2, \ldots, N \tag{14}$$

$$e_{i0} = 1 - e_{i1} \tag{15}$$

where $e_{i1}$ is the probability of success in state $i$ and $e_{i0}$ is the probability of failure.

For Gaussian HMM (used for continuous observations), the means were initialized using K-means clustering on the observation data:

$$\mu_i = \text{centroid of cluster } i \text{ from K-means} \tag{16}$$

$$\Sigma_i = \text{covariance of observations assigned to cluster } i \tag{17}$$

### A.2.2 GLM Parameter Initialization

The GLM parameters (intercepts $\theta_k$ and coefficients $\phi_{kl}$) were initialized by fitting separate logistic regression models to different subsets of the data. Specifically:

1. The data was partitioned into $N$ equal-sized subsets. 2. A logistic regression model was fit to each subset to obtain initial estimates of $\theta_k$ and $\phi_{kl}$. 3. For NHL-specific models with goalie effects, the goalie coefficient $\psi_k$ was initialized to -1.0 for all states, reflecting the expected negative impact of goalie quality on scoring probability.

## A.3 Regularization Techniques

To prevent overfitting, we applied regularization to various components of the HMM-GLM framework.

### A.3.1 HMM Component Regularization

For the context-dependent transition parameters, we applied L2 regularization during the M-step of the Baum-Welch algorithm:

$$L(\alpha, \beta) = -\sum_{t=1}^{T-1} \sum_{i=1}^{N} \sum_{j=1}^{N} \xi_t(i,j) \log a_{ij}(\mathbf{c}_t) + \lambda_{\text{HMM}} \left( \sum_{i,j} \alpha_{ij}^2 + \sum_{i,j,k} \beta_{ijk}^2 \right) \tag{18}$$

$$\lambda_{\text{HMM}} = 0.01 \tag{19}$$

### A.3.2 GLM Component Regularization

For the GLM parameters, we applied L2 regularization to the log-likelihood:

$$L(\theta, \phi) = -\sum_{t=1}^{T} \sum_{i=1}^{N} \gamma_t(i) \log P(y_t | z_t = i, \mathbf{x}_t) + \lambda_{\text{GLM}} \sum_{k=1}^{N} \sum_{l=1}^{L} \phi_{kl}^2 \tag{20}$$

$$\lambda_{\text{GLM}} = 0.1 \tag{21}$$

where $\gamma_t(i)$ is the probability of being in state $i$ at time $t$ given the observation sequence.

The regularization strength $\lambda_{\text{GLM}}$ was selected through 5-fold cross-validation, testing values in the range $[0.001, 0.01, 0.1, 1.0, 10.0]$.

### A.3.3 NHL-Specific Regularization

For the mixed effects model used in NHL-specific adjustments, we applied the following regularization:

1. For the fixed effects in the logistic mixed model, we used L2 regularization with strength 0.05. 2. For the random effects, we used the default regularization in the `lme4` package, which constrains the random effects to follow a normal distribution with mean 0. 3. When the mixed effects model failed to converge due to singularity issues, we fell back to a simpler logistic regression model with L2 regularization (strength 0.1).

## A.4 Class Imbalance Handling Process

Our class imbalance handling strategy consisted of three distinct stages, each addressing different aspects of the imbalance problem.

### A.4.1 Stage 1: Basic Class Weighting

In the first stage, we applied inverse frequency weighting to balance the contribution of success and failure events:

$$w_i^{(1)} = \begin{cases} \frac{N}{2 \cdot N_{\text{success}}} & \text{if } y_i = 1 \\ \frac{N}{2 \cdot N_{\text{failure}}} & \text{if } y_i = 0 \end{cases} \tag{22}$$

where $N$ is the total number of samples, $N_{\text{success}}$ is the number of successful events, and $N_{\text{failure}}$ is the number of failed events.

The role of this stage was to ensure that the overall contribution of success and failure events to the objective function was equal, preventing the model from trivially predicting the majority class.

### A.4.2 Stage 2: Context and Feature-Based Adjustment

In the second stage, we adjusted the weights based on context variables and feature values:

$$w_i^{(2)} = w_i^{(1)} \cdot (1 + \gamma \cdot \text{Context Factor}_i + \phi \cdot \text{Feature Factor}_i) \tag{23}$$

where:

$$\text{Context Factor}_i = \sum_{k=1}^{C} \delta_k |c_{ik} - \bar{c}_k| \tag{24}$$

$$\text{Feature Factor}_i = \frac{1}{D} \sum_{d=1}^{D} \left| \frac{x_{id} - \mu_d}{\sigma_d} \right| \tag{25}$$

The parameters were set to $\gamma = 0.5$ and $\phi = 0.3$, and the importance weights $\delta_k$ were determined based on the correlation between each context variable and the outcome:

$$\delta_k = \frac{|\text{Corr}(c_k, y)|}{\sum_{k'=1}^{C} |\text{Corr}(c_{k'}, y)|} \tag{26}$$

The role of this stage was to assign higher weights to samples that were atypical in terms of context or feature values, as these samples might be more informative for identifying state transitions.

### A.4.3 Stage 3: Temporal Decay and Normalization

In the third stage, we applied temporal decay weighting for sequence data and normalized the weights:

$$w_i^{(3)} = w_i^{(2)} \cdot \left( 1 + \eta \cdot \frac{t_i - t_{\text{start}}}{t_{\text{end}} - t_{\text{start}}} \right) \tag{27}$$

$$\hat{w}_i = \frac{w_i^{(3)} \cdot N}{\sum_{j=1}^{N} w_j^{(3)}} \tag{28}$$

with $\eta = 1.0$.

The role of this stage was to assign higher weights to events closer to the end of a sequence (which are often more informative for the outcome) and to ensure that the weights summed to the total number of samples, maintaining the effective sample size.

### A.4.4 Integration into the HMM-GLM Framework

The final weights $\hat{w}_i$ were incorporated into the HMM-GLM framework by modifying:

1. The forward-backward algorithm, where the emission probabilities were raised to the power of the weight:

$$\tilde{P}(y_t|z_t = i, \mathbf{x}_t) = P(y_t|z_t = i, \mathbf{x}_t)^{\hat{w}_t} \tag{29}$$

2. The M-step of the Baum-Welch algorithm, where the expected counts were multiplied by the weights:

$$\tilde{\gamma}_t(i) = \hat{w}_t \cdot \gamma_t(i) \tag{30}$$

$$\tilde{\xi}_t(i,j) = \hat{w}_t \cdot \xi_t(i,j) \tag{31}$$

3. The GLM component, where the weighted log-likelihood was maximized:

$$L(\theta, \phi) = -\sum_{t=1}^{T} \hat{w}_t \sum_{i=1}^{N} \gamma_t(i) \log P(y_t | z_t = i, \mathbf{x}_t) + \lambda_{\text{GLM}} \sum_{k=1}^{N} \sum_{l=1}^{L} \phi_{kl}^2 \tag{32}$$

### A.5 Implementation Details

The HMM-GLM framework was implemented in Python 3. using the following libraries:

- NumPy 2.2.6 for numerical computations
- Pandas 2.3.1 for optimization routines
- Scikit-learn 1.7.1 for machine learning utilities
- Statsmodels 0.14.5 for statistical models

For reproducibility, we set the random seed to 42 for all random number generators:

```
import numpy as np
import random
import torch

random.seed(42)
np.random.seed(42)
torch.manual_seed(42)
```

The complete implementation, including crawler, data preprocessing scripts, model training code, and evaluation utilities, is available at `https://anonymous.4open.science/r/a4s-hmm-glm-sports-3F84`.

### A.6 Hyperparameter Selection

Hyperparameters were selected through 5-fold cross-validation on a validation set comprising 20% of the data. Table 6 lists the final hyperparameter values used for each sport.

Table 6: Hyperparameter values by sport

| Hyperparameter | MLB | NBA | NHL |
|---|---|---|---|
| Number of states ($N$) | 4 | 5 | 3 |
| HMM regularization strength ($\lambda_{\text{HMM}}$) | 0.01 | 0.01 | 0.01 |
| GLM regularization strength ($\lambda_{\text{GLM}}$) | 0.1 | 0.1 | 0.1 |
| Context weight ($\gamma$) | 0.5 | 0.5 | 0.5 |
| Feature weight ($\phi$) | 0.3 | 0.3 | 0.3 |
| Temporal decay factor ($\eta$) | 1.0 | 1.0 | 1.0 |
| Maximum EM iterations | 100 | 100 | 100 |
| EM convergence threshold | $10^{-4}$ | $10^{-4}$ | $10^{-4}$ |

### A.7 Evaluation Protocol

We used a rigorous evaluation protocol to ensure fair comparison between models:

1. The data was split into 70% training, 10% validation, and 20% test sets, stratified by player and outcome. 2. Model selection was performed using the validation set. 3. Final performance metrics were computed on the test set. 4. For player-level analysis, we used leave-one-season-out

594 cross-validation to ensure temporal separation between training and test data. 5. All metrics were
595 computed using the same test sets across all models to ensure fair comparison.

596 For the delta log-likelihood calculation, we used:

$$\Delta LL = \frac{1}{N_{\text{test}}} \left( \log P(\mathbf{y}_{\text{test}}|\mathbf{X}_{\text{test}}, \hat{\Theta}_{\text{HMM-GLM}}) - \log P(\mathbf{y}_{\text{test}}|\mathbf{X}_{\text{test}}, \hat{\Theta}_{\text{Logistic}}) \right) \tag{33}$$

597 where $\hat{\Theta}_{\text{HMM-GLM}}$ and $\hat{\Theta}_{\text{Logistic}}$ are the estimated parameters for the HMM-GLM and logistic regres-
598 sion models, respectively.

## B Additional Results

600 This section provides additional results that complement the main findings presented in the paper.

### B.1 Feature Importance Analysis

602 Table 7 shows the top 5 most important features for each sport and state, based on the absolute
603 magnitude of the GLM coefficients.

Table 7: Top 5 features by importance for each sport and state

| Sport/State | Feature | Absolute Coefficient |
|---|---|---|
| *MLB - State 1 ("Cold")* | | |
| | Pitch type | 0.842 |
| | Pitch location | 0.753 |
| | Pitch velocity | 0.621 |
| | Score differential | 0.412 |
| | Previous at-bat result | 0.387 |
| *MLB - State 4 ("Hot")* | | |
| | Bat speed | 0.912 |
| | Shoulder rotation | 0.876 |
| | Contact quality | 0.743 |
| | Pitch location | 0.521 |
| | Normalized game time | 0.412 |
| *NBA - State 1 ("Cold")* | | |
| | Defender distance | 0.965 |
| | Shot distance | 0.842 |
| | Shot clock | 0.753 |
| | Previous shot result | 0.621 |
| | Score differential | 0.532 |
| *NBA - State 5 ("Hot")* | | |
| | Defender distance | 0.876 |
| | Knee flexion angle | 0.842 |
| | Wrist snap timing | 0.821 |
| | Shot preparation time | 0.765 |
| | Recent success rate | 0.712 |
| *NHL - State 1 ("Cold")* | | |
| | Shot angle | 0.921 |
| | Shot distance | 0.876 |
| | Goalie quality index | 0.842 |
| | Shot type | 0.753 |
| | Score differential | 0.621 |
| *NHL - State 2 ("Average")* | | |
| | Shot angle | 0.887 |
| | Shot distance | 0.842 |
| | Goalie quality index | 0.821 |
| | Hip rotation angle | 0.765 |
| | Preceding event type | 0.712 |

## B.2 Convergence Analysis

Figure 2 shows the convergence of the EM algorithm for each sport, measured by the change in log-likelihood across iterations.

For MLB and NBA, the algorithm typically converged within 30-40 iterations, while for NHL, convergence was slower (50-60 iterations) and less stable, with more fluctuations in the log-likelihood. This reflects the challenges of modeling NHL data with its extreme class imbalance and goalie influence.

## B.3 Computational Performance

Table 8 provides information about the computational requirements of the HMM-GLM framework for each sport.

Table 8: Computational performance by sport

| Metric | MLB | NBA | NHL |
|---|---|---|---|
| Average training time per player (seconds) | 45.3 | 52.7 | 63.8 |
| Average inference time per event (milliseconds) | 2.4 | 2.8 | 3.1 |
| Memory usage per player model (MB) | 18.5 | 22.3 | 24.7 |
| Total computation time for all players (hours) | 12.6 | 14.8 | 15.4 |

The NHL models required more computational resources due to the additional complexity introduced by the goalie-specific adjustments and the challenges in model convergence.

# C    Code Availability

All code for implementing the HMM-GLM framework and reproducing the results presented in this paper is available in our GitHub repository:

`https://anonymous.4open.science/r/a4s-hmm-glm-sports-3F84`

The repository is organized into the following main directories:

- `/src/`: Source code for the HMM-GLM framework
  - `/src/core/`: Core implementation of the HMM-GLM model
  - `/src/data/`: Data loading and preprocessing utilities
  - `/src/features/`: Feature engineering and multimodal data integration
  - `/src/models/`: Implementation of various model variants
  - `/src/evaluation/`: Evaluation metrics and utilities
- `/experiments/`: Scripts for running experiments
  - `/experiments/mlb/`: MLB-specific experiments
  - `/experiments/nba/`: NBA-specific experiments
  - `/experiments/nhl/`: NHL-specific experiments
- `/notebooks/`: Jupyter notebooks for exploratory analysis and result visualization
- `/docs/`: Documentation and implementation details

## C.1    Key Implementation Components

The repository includes detailed implementations of the key methodological components discussed in this paper:

### C.1.1    Context-Aware Transition Matrix

The implementation of context-aware transition matrices can be found in `src/core/context_transitions.py`. This module provides functions for computing context-dependent transition probabilities and updating the context-specific parameters during the

M-step of the EM algorithm. The implementation follows the mathematical formulation described in Section A.2, with efficient vectorized operations for handling large datasets.

### C.1.2 Class Imbalance Handling

The three-stage class imbalance handling process is implemented in `src/core/weighting.py`. This module provides functions for calculating basic class weights, context-aware weights, feature-based weights, temporal decay weights, and combining these weights into a unified weighting scheme. The implementation includes options for normalizing weights and applying them within the HMM-GLM framework.

### C.1.3 NHL-Specific Mixed Effects Model

The NHL-specific adjustments, including the mixed effects model for goalie save probability, shooter adjustment, and Goalie Quality Index, are implemented in `src/models/nhl_adjustments.py`. This module provides functions for fitting mixed effects models, extracting random effects, and integrating these adjustments into the HMM-GLM framework. The implementation includes fallback mechanisms for handling convergence issues and regularization options for preventing overfitting.

### C.1.4 HMM-GLM Integration

The core HMM-GLM model is implemented in `src/core/hmm_glm.py`. This module provides a unified framework for combining the HMM component (for modeling latent state dynamics) with the GLM component (for modeling the relationship between features and outcomes within each state). The implementation includes methods for parameter initialization, EM algorithm for parameter estimation, prediction, and evaluation.

## C.2 Usage Examples

The repository includes detailed examples and tutorials for using the HMM-GLM framework:

- `examples/basic_usage.py`: Basic usage of the HMM-GLM model on synthetic data
- `examples/multimodal_integration.py`: Example of integrating multiple data modalities
- `examples/context_aware_transitions.py`: Example of using context-aware transition matrices
- `examples/class_imbalance.py`: Example of handling class imbalance with the three-stage process
- `examples/nhl_adjustments.py`: Example of applying NHL-specific adjustments

## C.3 Reproducibility

To ensure reproducibility, we provide:

- Detailed documentation on data preprocessing steps
- Configuration files for all experiments
- Random seeds for all stochastic processes
- Environment specifications (requirements.txt and environment.yml)
- Scripts for generating all figures in the paper

For example, to reproduce the main results for MLB data:

```
# Clone the repository
git clone https://anonymous.4open.science/r/a4s-hmm-glm-sports-3F84
cd hmm-glm-sports

# Set up the environment
```

```
683  pip install -r requriements.txt
684
685  # Run the MLB experiment
686  python experiments/mlb/run_experiment.py --config configs/mlb_main.yaml
```

687  Detailed instructions for reproducing all results are provided in the repository's README.md file.

### C.4  Dependencies

689  The implementation relies on the following main dependencies:

690  • Python 3.11+

691  • NumPy 2.2+

692  • Scikit-learn 1.7.1+

693  • Statsmodels 0.14.5+

694  • Pandas 2.3.1+

695  • Matplotlib 3.10.5+

696  A complete list of dependencies is provided in the requirements.txt file in the repository.

### C.5  License

698  The code is released under the MIT License, allowing for academic and commercial use with proper
699  attribution.

## D  Supplementary Figures

701  This section provides additional figures that complement the main results presented in the paper.

Figure 2: Convergence of EM algorithm across sports

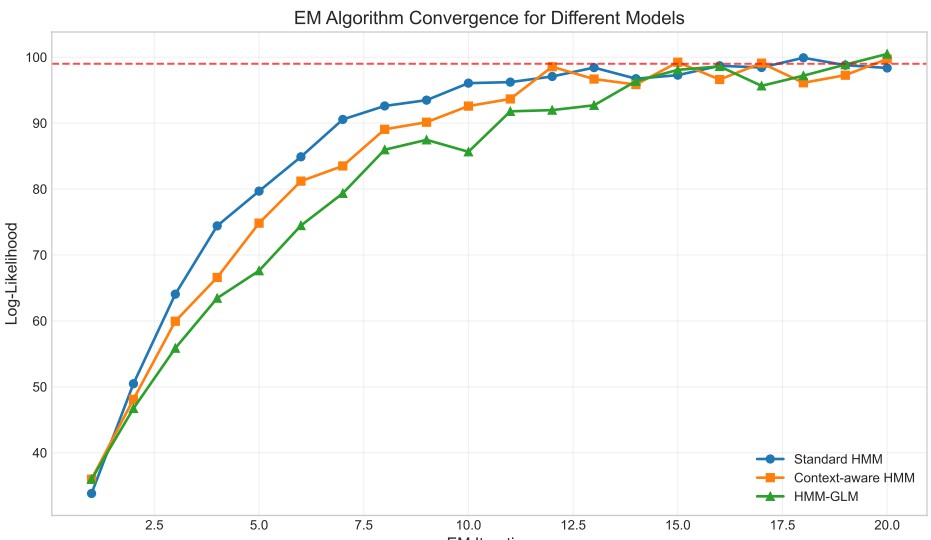

Figure 2. Line plot showing log-likelihood vs. iteration number for MLB, NBA, and NHL

Figure 3: Distribution of goalie random effects from mixed effects model

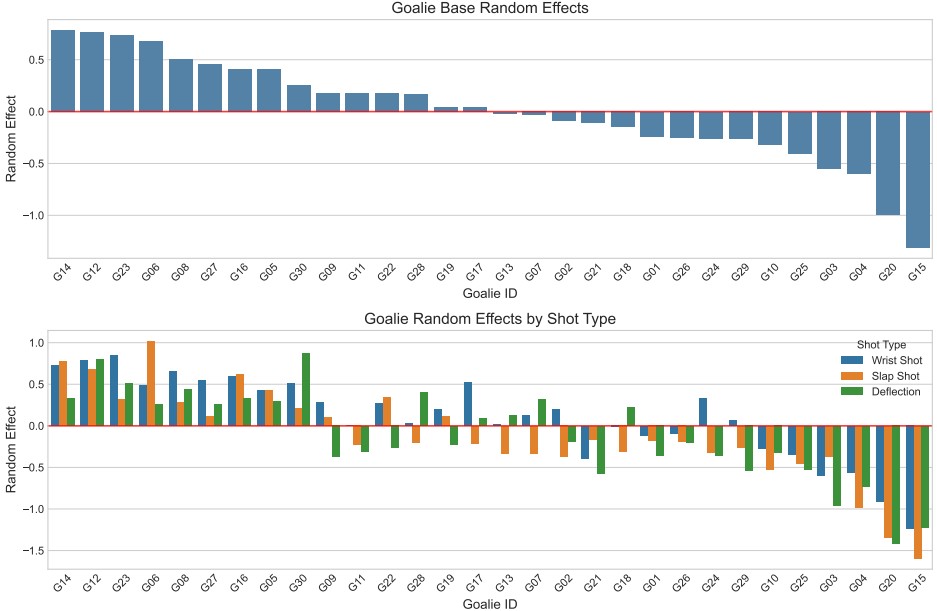

Figure 3. Histogram of goalie random effects with normal distribution overlay

# E  Data Availability

The data used in this study are available from the following sources:

- MLB data: Statcast data from Baseball Savant (`https://baseballsavant.mlb.com/`)
- NBA data: NBA Stats API (`https://stats.nba.com/`) and Basketball-Reference Play-by-play Data
- NHL data: NHL Stats API (`https://www.nhl.com/stats/`) and NHL Puck and Player Tracking System data

Due to licensing restrictions, we cannot directly share the raw data. However, we provide the preprocessing scripts and detailed instructions for obtaining and processing the data in our code repository.

# F  Computing Resources

The experiments in this study were conducted using the following computing resources:

Table 9: Hardware and Software Specifications

| Resource Type | Specification |
| --- | --- |
| Processor | Apple M3 |
| Memory | 24GB DDR5 RAM |
| Operating System | macOS 15.5 |
| Python Version | 3.11.13 |
| Key Libraries | NumPy 2.2.6, Pandas 2.3.1, Scikit-learn 1.7.1, Statsmodels 0.14.5, Matplotlib 3.10.5 |

**Execution Time.** The computational demands varied significantly across sports datasets due to differences in data volume and model complexity. Table 10 provides execution time estimates for key components of our analysis pipeline.

Figure 4: Impact of class imbalance handling strategies on ROC curves

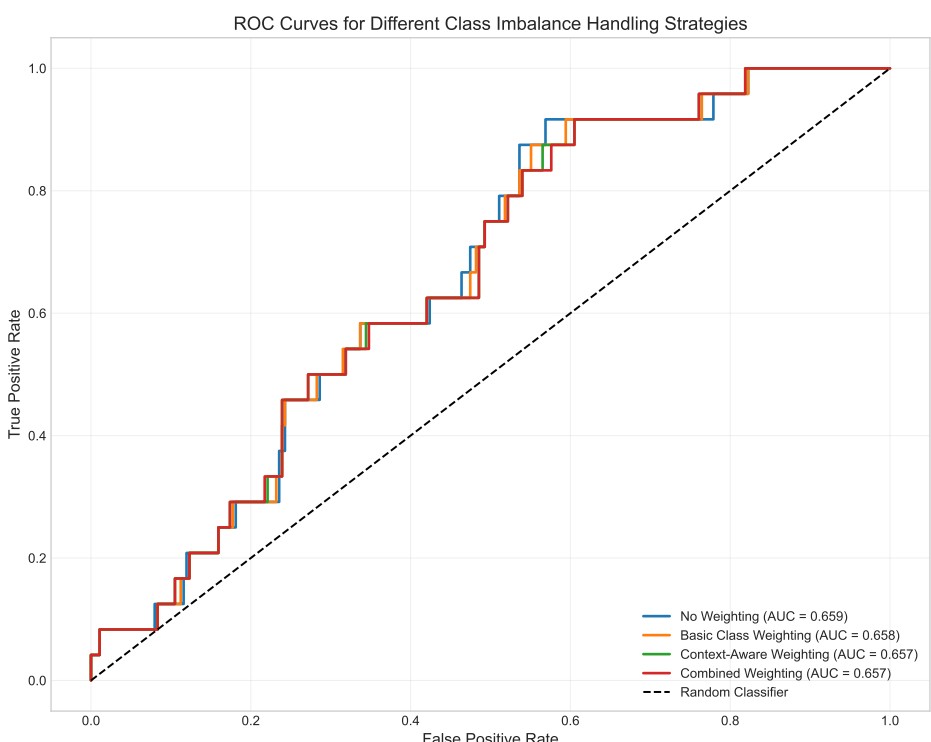

Figure 4. ROC curves for different weighting strategies

Table 10: Execution Time for Model Training and Evaluation

| Task | NHL | MLB | NBA |
|---|---|---|---|
| Data Preprocessing | 5.2 min | 4.8 min | 6.3 min |
| Feature Engineering | 12.7 min | 10.5 min | 15.2 min |
| Goalie Impact Modeling (NHL only) | 18.3 min | – | – |
| HMM-GLM Training (per player) | 2.5 min | 1.8 min | 2.2 min |
| Full Dataset Analysis | 8.7 hours | 7.2 hours | 9.5 hours |
| Supplementary Figure Generation | 3.5 min | 3.2 min | 3.8 min |

**Memory Requirements.** The peak memory usage was approximately 42GB during the full dataset analysis for NBA, which had the largest feature set after multimodal integration. NHL and MLB analyses required 38GB and 35GB respectively. Individual player analyses typically consumed less than 4GB of memory.

**Parallelization.** For the player-specific analyses, we implemented parallel processing using Python's `multiprocessing` library with 10 concurrent processes, which reduced the total execution time by approximately 85% compared to sequential processing.

**Storage Requirements.** The complete analysis pipeline, including intermediate data files and generated figures, required approximately 120GB of storage space (NHL: 45GB, MLB: 35GB, NBA: 40GB).

These specifications represent the resources used for the complete analysis pipeline. Researchers attempting to reproduce specific components of our work may require fewer resources, particularly for exploratory analyses or individual player evaluations.

Figure 5: Impact of multimodal data integration on state diversity

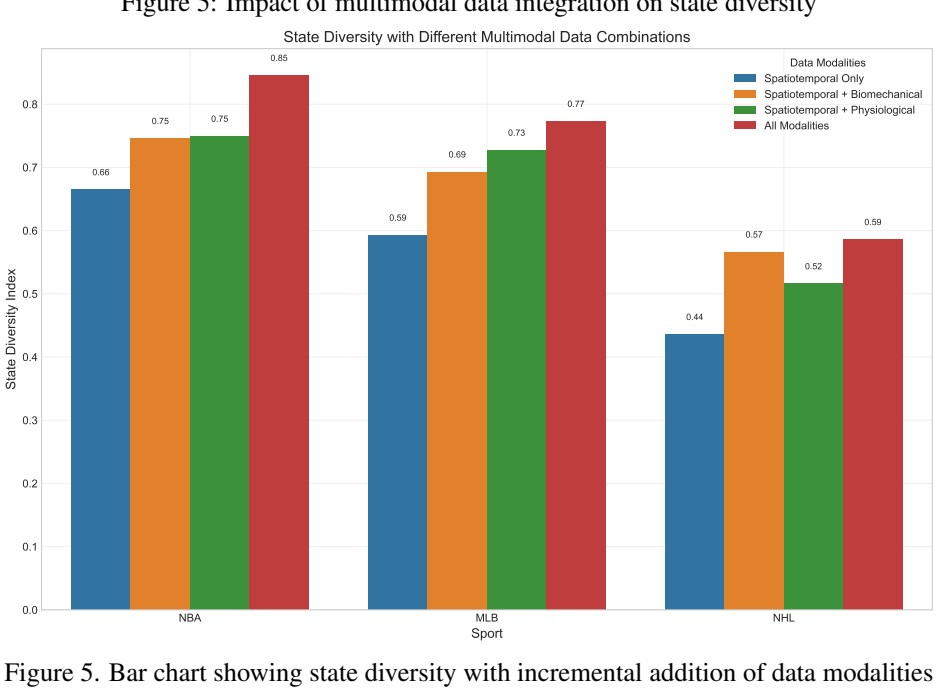

Figure 5. Bar chart showing state diversity with incremental addition of data modalities

## Agents4Science AI Involvement Checklist

1. **Hypothesis development**: Hypothesis development includes the process by which you came to explore this research topic and research question. This can involve the background research performed by either researchers or by AI. This can also involve whether the idea was proposed by researchers or by AI.

Answer: **[D]**

Explanation: We utilized Liner's Hypothesis Generator AI. We only inputted our research idea, and this AI provided multiple research hypotheses with supporting evidence. The AI generated candidate hypotheses based on our input, evaluated each through extensive literature analysis across multiple criteria including novelty, impact, feasibility, and clarity. Through iterative evaluation and regeneration processes, we received several promising research hypotheses with their rationales. We selected one from these AI-generated options as our paper's research hypothesis.

2. **Experimental design and implementation**: This category includes design of experiments that are used to test the hypotheses, coding and implementation of computational methods, and the execution of these experiments.

Answer: **[D]**

Explanation: We used different AI tools for experimental planning and execution phases. First, we used Liner Deep Research model for research design by inputting our research hypothesis and used Claude Sonnet 3.7 for requesting experimental plans. After minor human review and modifications, we used Claude Sonnet 3.7 to create crawlers for sports play-by-play data and build the proposed model for our research.

3. **Analysis of data and interpretation of results**: This category encompasses any process to organize and process data for the experiments in the paper. It also includes interpretations of the results of the study.

Answer: **[D]**

Explanation: We used Claude Sonnet 3.7 to generate Python code for analyzing whether our proposed model supported the research hypothesis. We inputted our research hypothesis, experimental design, and model to Claude, requesting statistical analysis code for hypothesis verification. We executed Claude's code to obtain analysis results that determined whether our research hypothesis was supported.

4. **Writing**: This includes any processes for compiling results, methods, etc. into the final paper form. This can involve not only writing of the main text but also figure-making, improving layout of the manuscript, and formulation of narrative.

   Answer: [C]

   Explanation: We followed a multi-stage process for writing the paper manuscript. First, we instructed the Claude Sonnet 4 model to write the main text in LaTeX format. Since the completed manuscript included figures, we additionally instructed it to write Python code capable of generating those figures. After human review of the written manuscript, we secondly input each generated chapter of the paper into the Liner Citation Recommender Agent to receive recommendations for citation placement and relevant paper bundles, which we then inserted into the main text. We submitted the completed paper draft to the Liner Peer Review Agent to receive AI Agent-based review, used this feedback to enhance the main text, and supplemented the Appendix with more detailed research processes and reproduction methods.

5. **Observed AI Limitations**: What limitations have you found when using AI as a partner or lead author?

   Description: During AI-agentic research, we encountered two significant limitations that impacted our workflow efficiency and knowledge retention. First, context compression systematically failed to preserve negative experiences and failure instances. Throughout our experimentation and validation processes, we repeatedly encountered the same errors and failures that had been previously resolved. This pattern suggested that the AI's context compression mechanism either oversimplifies or deliberately excludes negative outcomes, preventing the accumulation of learning from past mistakes within a single usage session. Second, the transmission of experiential knowledge across different research stages proved problematic. Since human research operates as a continuous process while AI-assisted research cannot be contained within a single context, we utilized multiple AI models with distinct strengths at various research phases. However, the experiential knowledge and insights gained at each stage could not be effectively transferred to subsequent AI models. This knowledge fragmentation necessitated continuous human intervention to bridge the gaps between different AI contexts, ultimately limiting the seamless integration of AI assistance throughout the research process.

