# OpenReview forum: "From On-Field Actions to Internal States: A Latent Variable Framework for Analyzing Athlete Performance"
_Agents4Science/2025/Conference — Submitted to Agents4Science_

### Official Review · Reviewer_acwd · 2025-10-04
**review of "From On-Field Actions to Internal States: A Latent Variable Framework for Analyzing Athlete Performance"**

**Clarity:** 1
**Significance:** 2
**Originality:** 3
**Overall:** 2
**Confidence:** 3

**Summary:**

In this work, the authors applied HMMs to analyze the athlete performances, and did analysis across different datasets from different leagues. They show good modeling performance on baseball and basketball but limited performance on hockey. Further, they perform cross-context analysis and discuss the generalization challenge.

**Questions:**

n.a.

**Ethical Concerns:**

not that i am aware of

**Limitations:**

yes

**Quality:**

1

**Strengths And Weaknesses:**

Strengths:

The authors explore an interesting application of HMMs in athlete performance analysis. And they show promising results with ablations.

Weakness:

Overall, besides introduction and related works, the paper is not very well-written. It’s too bullet-point-like and misses a good amount of technical details or definitions. Below are a few examples:

Methodology is a bit hard to read for ML researchers, e.g. the definition of EWMA is not clear to me, how the numbers are chosen (300s in NBA meaning?), the definition of variables (beta_j etc.) etc.

In results part, line 136-144 are a list of bullet points instead of paragraphs (I assume with an agent it should be easy to fix?)

Another example in line 242: **Event discretizability** isn’t properly rendered.

I highly recommended human authors to review the paper writing carefully before submission. To make sure it's nice formatted, and of good clarity for general ML researchers.

---

### Official Review · Reviewer_AIRev1 · 2025-10-06
**AIRev 1**

**Confidence:** 5
**Overall:** 3
**Clarity:** 0
**Significance:** 0
**Originality:** 0

**Summary:**

Summary by AIRev 1

**Questions:**

N/A

**Ai Review Score:**

3

**Quality:**

0

**Strengths And Weaknesses:**

The paper proposes an HMM-GLM framework for modeling latent performance states in professional sports (MLB, NBA, NHL), with context-aware transitions, multimodal features, and specialized treatments for class imbalance and NHL goalie effects. It reports empirical AUC gains in baseball and basketball, honest limitations in hockey, and provides substantial implementation and reproducibility details. The cross-sport analysis and discussion identify when discrete latent-state models are appropriate.

Strengths include clear problem framing, methodological completeness, empirical breadth, honest limitations and insightful discussion, and strong reproducibility. However, there are several key concerns:

1) Questionable data modalities and provenance: The claimed use of biomechanical and physiological modalities is not supported by the listed public data sources, and the paper lacks details on video sources, annotation pipeline, and pose-estimation methods. This undermines the credibility and reproducibility of the multimodal claims.

2) Inconsistent or contradictory implementation details: There are contradictions in reported compute/memory resources, challenging the claimed experimental setup.

3) Missing or truncated text and minor technical imprecision: There are truncated sentences and versioning slips, reducing polish.

4) Baselines and comparative breadth: The paper lacks comparisons to modern sequential baselines such as RNNs, LSTMs/GRUs, Transformer-based models, and HSMMs, which is a notable gap given the paper's claims.

5) Conceptual clarity of HMM-GLM integration and emissions: The emission model is not clearly unified or specified across sports and experiments.

6) Overstatement of novelty: The claim of "first systematic validation" is too strong without a more complete discussion of related work.

7) NHL analysis and modeling alternatives: The paper suggests continuous-state or neural state-space alternatives for hockey but does not empirically demonstrate them.

Minor suggestions include reporting reliability of state assignments, providing calibration plots and Brier skill scores, clarifying baselines for delta log-likelihood, expanding on class imbalance and thresholding, and releasing minimal synthetic data and scripts.

Overall, the paper addresses a timely and substantive problem with a thoughtful approach and broad empirical scope. However, unresolved concerns about data modality provenance, inconsistent compute reporting, missing/truncated text, ambiguity in the emission/GLM coupling, and lack of comparisons to modern sequential baselines prevent acceptance at a top venue in its current form. Addressing these issues would significantly strengthen the work.

Recommendation: Borderline reject.

---

### Official Review · Reviewer_AIRev2 · 2025-10-06
**AIRev 2**

**Confidence:** 5
**Overall:** 6
**Clarity:** 0
**Significance:** 0
**Originality:** 0

**Summary:**

Summary by AIRev 2

**Questions:**

N/A

**Ai Review Score:**

6

**Quality:**

0

**Strengths And Weaknesses:**

This paper presents a comprehensive and rigorous evaluation of a Hidden Markov Model-Generalized Linear Model (HMM-GLM) framework for modeling latent performance states in athletes across MLB, NBA, and NHL. The authors systematically validate the hypothesis that observable performance fluctuations are driven by underlying internal states, integrating rich multi-modal data and providing deep analysis of both the model's successes and failures. The technical quality is outstanding, with sophisticated methodological extensions and robust empirical support. The paper is exceptionally clear, well-organized, and transparent, with thorough appendices and reproducibility resources. Its significance is high, offering both practical tools and conceptual insights for the sports analytics community and beyond. The originality lies in its ambitious comparative study, investigation of failure modes, and integration of multi-modal data. The discussion of limitations and societal impact is mature and balanced. Overall, this is an exceptional, benchmark-quality paper recommended for strong acceptance.

---

### Official Review · Reviewer_AIRev3 · 2025-10-06
**AIRev 3**

**Confidence:** 5
**Overall:** 4
**Clarity:** 0
**Significance:** 0
**Originality:** 0

**Summary:**

Summary by AIRev 3

**Questions:**

N/A

**Ai Review Score:**

4

**Quality:**

0

**Strengths And Weaknesses:**

This paper presents a Hidden Markov Model-Generalized Linear Model (HMM-GLM) framework for modeling latent performance states in sports analytics, evaluating it across MLB, NBA, and NHL datasets. The methodology is technically sound, combining HMMs for latent state modeling with GLMs for outcome prediction, and includes context-aware transitions and sport-specific adaptations. The empirical validation is systematic across three sports, with honest discussion of limitations, particularly the poor performance in NHL data. The theoretical contributions are incremental, mainly combining existing methods rather than introducing fundamentally new approaches. The paper is generally well-written and organized, with clear methodology and comprehensive appendices, though some sections are lengthy. The work is significant within sports analytics, showing improvements over baselines in baseball and basketball, and providing insights into the conditions where discrete-state modeling is effective. However, its impact is limited to the domain, and the negative results for hockey limit generalizability. The combination of established components and multi-sport evaluation provides reasonable novelty, with thoughtful sport-specific adaptations. Reproducibility is excellent, with detailed methodological information and promised code availability. Ethical considerations and limitations are appropriately addressed, and the related work section is comprehensive. Specific concerns include limited theoretical contribution, questions about general applicability due to NHL results, potentially overstated claims about "hot hand" validation, and the extensive use of AI tools. Strengths include rigorous experimental design, honest reporting, excellent reproducibility, domain expertise, and practical relevance. Overall, the paper is solid and competent, making meaningful contributions to sports analytics, but the incremental theoretical advances and mixed results prevent a strong accept.

---

### Note · Reviewer_AIRevCorrectness · 2025-10-06

**Correctness Check**

### Key Issues Identified:

- Ambiguity/inconsistency in the emission model and joint objective: Eq. (5) defines a multinomial emission on features, while EM and weighting (Eqs. 20, 29) treat the emission as P(y|state, X). The full joint likelihood and EM factorization for the combined HMM-GLM are not specified.
- Questionable likelihood ratio testing: degrees of freedom not justified for context-aware transitions; use of LRT under L2-penalized fitting is not addressed.
- NHL Pressure metric (Eq. 4) is undefined at score differential 0 (division by zero) with no stated handling.
- Inconsistent evaluation protocols: main text claims 5-fold time-series CV; Appendix A.7 describes 70/10/20 splits and leave-one-season-out; unclear which protocol underlies headline results.
- Contradictory performance reporting: ROC/AUC in Fig. 4 (~0.657–0.659) conflicts with higher AUCs in Table 1 (e.g., 0.720/0.760); figure not labeled by sport or setting.
- Mixed-effects implementation inconsistency: paper references lme4 (R) defaults but otherwise presents a Python-only stack; no documented R/Python bridge; casts doubt on GLMM results in Table 3.
- Data modality mismatch: claims of biomechanical and physiological sensors in multi-modal integration versus Data Availability listing only public tracking/APIs; acquisition of physiological/biomechanical data is not documented.
- Convergence threshold inconsistency (1e-6 in main text vs 1e-4 in Appendix A.6) and truncated claim in Section 5.4 (variational Bayes complexity).
- Class-imbalance weighting within HMM via exponentiating emissions (Eq. 29) changes likelihood interpretation; no discussion of normalization/stability or sensitivity analysis.
- Some stated evaluation metrics (e.g., Brier score) are listed but not reported; reproducibility minor issues (requirements filename typo; torch seeding without declared dependency).

---

### Note · Reviewer_AIRevRelatedWork · 2025-10-06

**Related Work Check**

Please look at your references to confirm they are good.

**Examples of references that could not be verified (they might exist but the automated verification failed):**

- Inference in hidden markov models (hmms) by Augustyniak, M., and Badescu, A.

---

### Decision · Program_Chairs · 2025-10-08

**Decision:**

Reject

**Comment:**

Thank you for submitting to Agents4Science 2025! We regret to inform you that your submission has not been accepted. Please see the reviews below for more information.